# The Correlation of In Vivo MR Spectroscopy and Ex Vivo 2-Hydroxyglutarate Concentration for the Prediction of Isocitrate Dehydrogenase Mutation Status in Diffuse Glioma

**DOI:** 10.3390/diagnostics13172791

**Published:** 2023-08-29

**Authors:** Bart R. J. van Dijken, Hanne-Rinck Jeltema, Justyna Kłos, Peter Jan van Laar, Roelien H. Enting, Ronald G. H. J. Maatman, Klaas Bijsterveld, Wilfred F. A. Den Dunnen, Rudi A. Dierckx, Paul E. Sijens, Anouk van der Hoorn

**Affiliations:** 1Department of Radiology, Medical Imaging Center (MIC), University Medical Center Groningen (UMCG), 9700 RB Groningen, The Netherlands; 2Department of Neurosurgery, University Medical Center Groningen (UMCG), 9700 RB Groningen, The Netherlands; 3Department of Radiology, Hospital Group Twente (ZGT), 7600 SZ Almelo, The Netherlands; 4Department of Neurology, University Medical Center Groningen (UMCG), 9700 RB Groningen, The Netherlands; 5Department of Laboratory Medicine, University Medical Center Groningen (UMCG), 9700 RB Groningen, The Netherlands; 6Department of Pathology, University Medical Center Groningen (UMCG), 9700 RB Groningen, The Netherlands; 7Department of Nuclear Medicine, Medical Imaging Center (MIC), University Medical Center Groningen (UMCG), 9700 RB Groningen, The Netherlands

**Keywords:** glioma, IDH mutant glioma, MR spectroscopy, biopsy

## Abstract

Isocitrate dehydrogenase (IDH) mutation status is an important biomarker in the glioma-defining subtype and corresponding prognosis. This study proposes a straightforward method for 2-hydroxyglutarate (2-HG) quantification by MR spectroscopy for IDH mutation status detection and directly compares in vivo 2-HG MR spectroscopy with ex vivo 2-HG concentration measured in resected tumor tissue. Eleven patients with suspected lower-grade glioma (ten IDH1; one IDHwt) were prospectively included. Preoperatively, 3T point-resolved spectroscopy (PRESS) was acquired; 2-HG was measured as the percentage elevation of Glx3 (the sum of 2-HG and Glx) compared to Glx4. IDH mutation status was assessed by immunochemistry or direct sequencing. The ex vivo 2-HG concentration was determined in surgically obtained tissue specimens using gas chromatography–mass spectrometry. Pearson correlation was used for assessing the correlation between in vivo MR spectroscopy and ex vivo 2-HG concentration. MR spectroscopy was positive for 2-HG in eight patients, all of whom had IDH1 tumors. A strong correlation (r = 0.80, *p* = 0.003) between 2-HG MR spectroscopy and the ex vivo 2-HG concentration was found. This study shows in vivo 2-HG MR spectroscopy can non-invasively determine IDH status in glioma and demonstrates a strong correlation with ex vivo 2-HG concentration in patients with lower-grade glioma.

## 1. Introduction

Diffuse gliomas are a heterogeneous tumor population for which prognosis and treatment strategies vary according to the subtype [1]. The most recent guideline update centralizes molecular biomarkers, in addition to histological features, to categorize gliomas as they have been shown to be important prognostic markers [1,2,3]. Isocitrate dehydrogenase (IDH) mutation status is currently one of the most prominent biomarkers, most often occurring in World Health Organization (WHO) grades II and III gliomas (formerly known as lower-grade gliomas) [2,3]. Patients with IDH mutant gliomas have a prognosis of several years, with survival depending on histological features [1]. Because of this better prognosis, IDH mutant tumors with grade 4 histomorphology are now called astrocytoma, IDH mutant (WHO grade 4) [1].

Patients with tumors lacking IDH mutations, such as astrocytoma, IDH wild type (IDHwt) (WHO grade 2 or 3), and glioblastoma, IDHwt (WHO grade 4), are known to have a poorer prognosis than their IDH mutant counterparts [1,3]. Treatment strategies, therefore, differ between IDH mutant and IDHwt gliomas [3], with the latter being treated more aggressively [3,4]. In the former, it is sometimes justifiable to postpone adjuvant treatment after neurosurgical resection and adopt a ‘wait and see’ policy, particularly in patients with an IDH mutant glioma and favorable prognostic factors [3]. Reliable detection of IDH mutation status is, thus, fundamental, both for prognostication and establishing treatment decisions.

Definitive determination of IDH mutation status would be advantageous in several clinical situations. First, biopsy or resection could be delayed in stable patients with a tumor at an eloquent location and no imaging characteristics suspicious of a higher grade. Second, in cases where a wider differential diagnosis is considered. For example, it may be difficult to differentiate hippocampal lower-grade glioma from mesiotemporal sclerosis in its initial stage. Since the latter will not harbor an IDH1 mutation, a definitive diagnosis could be made through imaging alone when MR spectroscopy is positive.

MR spectroscopy is capable of detecting 2-hydroxyglutarate (2-HG) [5,6,7,8], an oncometabolite that is produced as a direct consequence of IDH mutations, measured at approximately 2.25 ppm [9]. Previous studies reported a high diagnostic accuracy of 2-HG MR spectroscopy for the prediction of IDH mutation status with a sensitivity and specificity of >90% [8]. However, previous studies did not directly measure the 2-HG concentration in the tumor to validate the (voxel-specific) spectroscopy results. Furthermore, large methodological differences exist between studies, such as single voxel versus multivoxel analysis, and a large heterogeneity in the echo times (TE) used. Thus, a standardized method for daily clinical practice is lacking [8].

In this prospective pilot study, we propose a straightforward method for the detection of 2-HG with MR spectroscopy and directly compare our imaging results with the ex vivo 2-HG concentration measured in resected tumor tissue. We hypothesized that in vivo 2-HG MR spectroscopy correlates well with the ex vivo 2-HG concentration in patients with diffuse glioma.

## 2. Methods

### 2.1. Patient Selection

Adult patients with a suspected lower-grade diffuse glioma, based on a non-enhancing lesion on MRI, were prospectively included in this study. Eligible patients had to be scheduled for maximal safe neurosurgical resection of the tumor and underwent preoperative MRI for neuronavigation purposes. Exclusion criteria were previous neurosurgery or radiotherapy to the brain and an inability to undergo MRI. Oligodendrogliomas were also excluded from the study. The study was approved by the local medical ethics committee (METc 2018/090), and written informed consent was obtained from all patients included.

### 2.2. Scanning Protocol

All patients included were scanned on the same 3T MRI scanner (Magnetom Prisma, Siemens Healthcare, Erlangen, Germany) according to our standard neuro-oncology protocol extended with MR spectroscopy. The neuro-oncology imaging protocol consisted of pre- and post-contrast 3D T1-weighted MPRage (repetition time (TR) 2300 ms, TE 2.32 ms, inversion time (TI) 900 ms, flip angle 8 degrees, slice thickness 0.9 mm, voxel size 0.9 × 0.9 × 0.9 mm); a transversal T2 susceptibility-weighted imaging (SWI) sequence (TR 28 ms, TE 20 ms, flip angle 15 degrees, slice thickness 2 mm, voxel size 0.7 × 0.7 × 2.0 mm); transversal diffusion-weighted imaging (DWI) (TR 440 ms, TEs 60 and 104 ms, flip angle 180 degrees, slice thickness 4 mm, voxel size 1.0 × 1.0 × 4.0 mm, with two b-values (0 and 1000 s/mm^2^); a 3D T2 fluid attenuated inversion recovery (FLAIR) sequence (TR 5000 ms, TE 391 ms, TI 1800 ms, slice thickness 1 mm, voxel size 1.0 × 1.0 × 1.0 mm); a transversal dynamic susceptibility contrast (DSC) perfusion sequence (TR 1780 ms, TE 30 ms, flip angle 90 degrees, slice thickness 4 mm, voxel size 0.9 × 0.9 × 4.0 mm); and a transversal T2 turbo spin echo (TSE) sequence (TR 8800 ms, TE 100 ms, flip angle 150 degrees, slice thickness 3 mm, voxel size 0.4 × 0.4 × 3.0 mm). An intravenous bolus of 20 cc gadolinium was given during DSC acquisition. A 64-channel head coil was used.

### 2.3. MR Spectroscopy Acquisition

MR spectroscopy was acquired prior to administration of contrast agent consisting of 2D-chemical shift imaging (CSI) point-resolved spectroscopy (PRESS) measurement with TR/TE = 1500/40 ms. Field of view was 16 × 16 voxels in typically a transverse plane and a voxel size of 10 × 10 × 20 mm^3^. The acquisition volume of interest (VOI) grid was placed over the tumor and contralateral hemisphere, using three pairs of saturation bands to minimize signals from outside the VOI, and the correct location was confirmed in three planes. The acquisition VOI was carefully placed within the brain parenchyma to avoid detection of lipids from the skull, including as much as possible of the entire lesion and contralateral hemisphere (Figure 1). Automated adjustments included localized multiple angle projection shimming, realizing water line widths (12 Hz) in the VOI and chemical shift excitation (CHESS), and spoiling of the resultant water signal, which yielded water suppression by a factor exceeding 10,000.

Standardized post-processing by manufacturer’s Syngo software (SyngoMMWP spectroscopy, version VE31H, Siemens Healthineers, Erlangen, Germany) consisted of water reference processing, Hanning filtering (center 0 ms, width 512 ms), zero filling from 1024 data points to 2048, Fourier transformation, frequency shift correction, sixth-order polynomial baseline correction, phase correction, and frequency domain curve fitting. The curve fitting was set to fit peaks to Gaussian line shapes, including the chemical shift ranges of 3.80–4.05 for Cr2, 3.62–3.70 for Ins2, 3.50–3.60 for Ins1, 2.14–3.34 ppm for Cho, 2.94–3.14 for Cr1, 2.25–2.35 for Glx3/2-HG, 2.31–2.40 for Glx4, and 1.82–2.22 for NAA. On the clinical MRI scanner, all of these data processing steps were fully automated, allowing for operator-independent quantification.

The 2-HG peak was measured as the percentage elevation of the glutamate–glutamine-3 (Glx3) peak, the sum of 2-HG and glutamate–glutamine, compared to the Glx4 peak, only consisting of glutamate–glutamine, relative to the Glx3/Glx4 ratio in uninvolved contralateral tissue (equal to 0.81 ± 0.06 SD amongst individuals and locations). Percentage of elevation >0% was considered positive.

### 2.4. Tissue Collection and Analysis

A maximum safe resection was performed on all subjects by an experienced oncological neurosurgeon (HRJ, 6 years of experience (*n* = 9)) and another oncological neurosurgeon with >15 years of experience (*n* = 2), guided by neuronavigation (Brainlab, Munich, Germany). Two tissue specimens were collected prior to resection of the tumor bulk. Specimens were immediately placed in liquid nitrogen. In two cases, only one sample was acquired; for one patient, four samples were taken. For five patients, the locations of the specimen collection were stored on the neuronavigation software (Cranial Navigation Application, Brainlab, Munich, Germany) and used for direct voxel sample comparison. Unfortunately, due to technical issues with the neuronavigation system, the biopsy locations for the other patients were not stored. The resected tumor was histologically assessed by a neuropathologist (WD, >15 years of experience). IDH mutation status was assessed by R132H immunochemistry. Direct IDH1 and IDH2 DNA sequencing were used for immunonegative cases to detect or exclude possible non-canonical mutations. The ex vivo 2-HG concentration was determined using gas chromatography–mass spectrometry and measured as 2-OH glutaric acid concentration in mmoles/kg tissue.

### 2.5. Statistics

Pearson correlation was used for testing correlation between MR spectroscopy (lowest, highest, and mean values) and ex vivo 2-HG concentration. A predetermined power calculation demonstrated a β of 0.75 for a correlation coefficient ≥0.70 based on a population of 10 subjects, which was deemed feasible for our study. Bootstrapping was performed to control for the relatively low number of included subjects. A two-sided *p*-value of 0.05 was used throughout this study.

## 3. Results

A total of 11 consecutive patients with a suspected lower-grade glioma planned for maximal surgery, seven men and four women, with a median age of 33 years (range 24–61), were included in this study. Nine patients had an IDH1 grade II astrocytoma. One patient had an IDH1 astrocytoma grade IV, demonstrating microvascular proliferation and necrosis. Seven patients had a tumor with the typical R132H mutation, whereas non-R132H mutations (R132C *n* = 2, R132S *n* = 1) were found in the three other patients with IDH1-mutated tumors. The remaining patient suffered from an IDHwt grade IV glioblastoma (see Table 1 for general characteristics).

MR spectroscopy was positive for 2-HG in eight patients, all of whom had IDH1 tumors. For two patients with a negative MR spectroscopy for 2-HG, histology did demonstrate an IDH1 mutation (patients 7 and 10 in Table 1). The other patient had an IDHwt tumor, corresponding with the negative MR spectroscopy. The positive predictive value (PPV) and negative predictive value (NPV) of 2-HG MR spectroscopy for our cohort were 100% and 33%, respectively, when a percentage elevation of Glx3 > 0% was used as a cut-off value.

Gas chromatography–mass spectrometry was able to correctly detect 2-HG in all IDH1-mutated tumors. The median ex vivo 2-HG concentration was 1.70 mmoles/kg tissue (IQR 0.82–3.25). Virtually no 2-HG (0.05 mmoles/kg) was detected in the IDHwt tumor specimen of patient 9. The 2-HG concentration was particularly low in one of the two IDH1 tumor patients with a negative MR spectroscopy (patient 7), namely 0.29 mmoles/kg. The tumor specimen of the other patient with a negative MR spectroscopy (patient 10) demonstrated a 2-HG concentration of 1.38 mmoles/kg.

The mean ex vivo 2-HG concentration was higher in non-R132H-mutated cases (5.45 mmoles/kg) than in cases with a typical R132H IDH mutation (1.38 mmoles/kg), *p* = 0.001. This was also the case for the ratios between the ex vivo 2-HG concentration and 2-HG spectroscopy (Table 1), with non-R132H cases demonstrating higher ratios (*p* = 0.035).

There was a strong correlation (*r* = 0.80, *p* = 0.003) between in vivo 2-HG MR spectroscopy and the mean ex vivo 2-HG concentration (Figure 2A). For the lowest and highest ex vivo 2-HG concentrations of the samples, this was r = 0.82, *p* = 0.002, and r = 0.80, *p* = 0.003, respectively. When excluding the three cases with a non-R132H mutation, the correlation did not change (r = 0.81, *p* = 0.014). For five patients, the location of specimen collection was stored on neuronavigation, and the tissue specimens could be compared with the corresponding voxel. We found a strong correlation (*r* = 0.83, *p* = 0.003) between the voxel-specific 2-HG MR spectroscopy peak and the ex vivo 2-HG concentration (Figure 2B and Figure 3).

## 4. Discussion

In this prospective study among 11 diffuse glioma patients, we evaluated 2-HG MR spectroscopy in relation to IDH mutation status and directly compared the MR spectroscopy results with the ex vivo 2-HG concentration measured within the resected tumor. We demonstrated that a strong correlation exists between 2-HG detected on MR spectroscopy and the ex vivo determined 2-HG concentration. As far as we know, this is the first study to directly compare the voxel-specific 2-HG value with the ex vivo 2-HG concentration of the tumor tissue samples, where we found an even stronger correlation.

IDH mutations occur in >80% of lower-grade (II/III) gliomas and approximately 5% of grade IV gliomas, now called astrocytoma, IDH mutant (WHO grade 4) [10,11]. Most common mutations are located in the IDH1 gene, located at R132, with a vast majority being the R132H subtype. Less frequently, other subtypes are found (R132C/R132G/R132S/R132L) [12]. In some patients with IDH-mutated gliomas, the IDH2 gene is affected, with usually R140 or R172 mutations [10]. IDH mutations alter the physiological conversion of isocitrate to α-ketoglutarate (α-KG) and lead to the formation of 2-HG [9,11]. Almost a decade ago, it was first demonstrated that 2-HG accumulation in IDH-mutated gliomas could be detected by MR spectroscopy [13]. J-coupled proton resonances of the two C4 protons of 2-HG are observed around 2.25 ppm, which heavily overlaps with the two C3 protons of glutamate and glutamine (Glx) and, therefore, potentially hinders reliable detection with MR spectroscopy [6,13]. The glutamate and glutamine C4 protons of around 2.35 ppm (Glx4) do not overlap with the C4 signals of 2-HG [14,15]. We thus introduced a straightforward method for 2-HG detection by comparing the Glx3 resonance, which is partly attributable to 2-HG, to Glx4. Considering that biopsies were taken as close to the skull as possible, we corrected the observed tumor Glx3/Glx4 ratios by division by those observed contralaterally.

Our MR spectroscopy results correctly predicted mutated IDH status in 80% of the cases. Moreover, we demonstrated a strong correlation between voxel-specific MR spectroscopy measured 2-HG and the ex vivo 2-HG concentration in resected tumor samples. Our results, therefore, add to a growing body of evidence that 2-HG MR spectroscopy is promising for non-invasively determining IDH status in glioma [8]. Recently, a second meta-analysis was published, which investigated the diagnostic accuracy of 2-HG MR spectroscopy for the determination of IDH status in subgroups of lower-grade gliomas and glioblastomas, respectively, which confirmed high sensitivity (93% (95%CI 58–99%)) and specificity (84% (51–96%)) for both cohorts [16]. However, a wide confidence interval was found due to the heterogeneity of the methodology of the included studies. There are several factors that have been shown to influence the diagnostic performance of 2-HG MR spectroscopy, such as TE, as demonstrated by Suh and colleagues [8,17]. It was found that long TEs led to a better diagnostic performance than shorter TEs. Furthermore, there is no consensus on the optimal cut-off value for 2-HG MR spectroscopy detection [8]. Our method avoids the usage of a predefined absolute cut-off value but rather measures the percentage elevation of Glx3 as compared to the Glx4 peak, making it more applicable to daily clinical practice since values >0% indicate a higher Glx3 concentration than Glx4 due to 2-HG. Considering that Hurd et al. [14] found respective glutamate and glutamine concentrations of 11.7 and 3.2 mmoles/kg in the gray matter and 7.1 and 1.7 in the white matter, the typical 15% increase in Glx3 observed here in IDH1 tumors (Figure 2) would amount to 0.15 × (11.7 + 3.2) = 2.2 mmoles/kg (GM) or 0.15 × (7.1 + 1.7) = 1.3 mmoles/kg (WM), assuming similar T1 and T2 relaxation properties for above-mentioned resonances of 2-HG and Glx. This appears to be quite in agreement with the median concentration ex vivo in the biopsies (1.73 mmoles/kg).

In our cohort, three out of ten patients had a rare non-R132H IDH1 mutation, which is more than expected. It is known that >90% of IDH1 mutant gliomas harbor an R132H mutation, with approximately 4% harboring an R132C mutation and the other subtypes occurring even less frequently [12]. The three non-R132H cases demonstrated a higher ex vivo 2-HG concentration than the R132H cases. However, when we excluded the non-R132H cases, our results did not change with a high correlation between 2-HG MR spectroscopy and the ex vivo 2HG concentration. The higher concentration of 2-HG for non-R132H subtypes has also been described by an earlier study [12]. This study by Pusch et al. hypothesized that only a moderately increased 2-HG concentration, as produced by R132H mutations, had beneficial proliferating effects and may explain the predominance of R132H cases [12]. The implications of different IDH1 mutation subtypes on the diagnostic performance of 2-HG MR spectroscopy should be further studied in larger cohorts.

## 5. Limitations

The most important limitation of our study is the small sample size, which limits the generalizability of our results. Despite our positive findings, two IDH1-mutated gliomas had false negative 2-HG MR spectroscopy results (20%), which is higher than previously described. The limited generalizability is also highlighted by the relatively large number of non-R132H mutations. Validation of our findings in a larger patient cohort is therefore necessary. Furthermore, biopsy location data used for direct voxel sample comparison were missing for approximately half of our included patients. Data could not be stored on the neuronavigation software due to technical issues with the neuronavigation system. Therefore, a voxel-specific MR spectroscopy comparison with the ex vivo 2-HG concentration within the biopsy tissue was available in only five patients (10 samples). However, this is the first study to directly compare the voxel-specific 2-HG value with the ex vivo 2-HG concentration of the tumor tissue samples, and we obtained a strong correlation between 2-HG MR spectroscopy measures and the ex vivo 2-HG concentration in the tissue.

## 6. Conclusions

This prospective pilot study is the first to directly compare in vivo 2-HG MR spectroscopy measures with the ex vivo 2-HG concentration measured in resected tumor tissue in patients with diffuse glioma. We proposed a straightforward method with a high diagnostic performance for 2-HG MR spectroscopy to non-invasively determine IDH mutation status in diffuse glioma. Despite the small number of patients included in this study, a strong correlation between 2-HG MR spectroscopy and ex vivo 2-HG concentration was found. Our results should prompt future studies to validate our findings in larger cohorts.

## Figures and Tables

**Figure 1 diagnostics-13-02791-f001:**
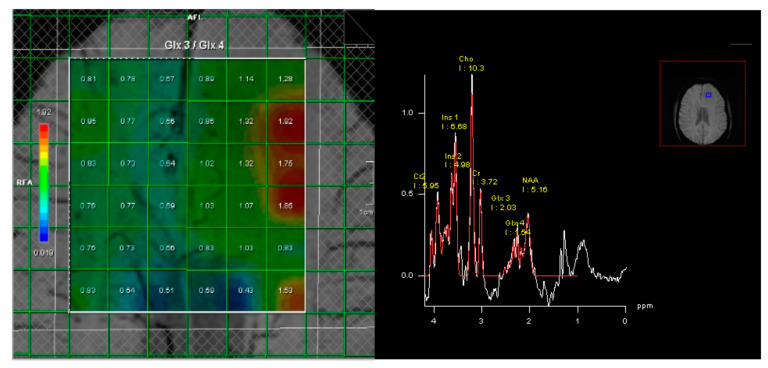
Case example of a positive 2-HG MR spectroscopy in a patient with a left frontal diffuse glioma. Immunochemistry demonstrated that the tumor was IDH1-mutated, as suggested by 2-HG spectroscopy. **Left**: A multivoxel MR spectroscopy grid is placed over the tumor and contralateral hemisphere in a transverse plane. Measures of Glx3/Glx4 are calculated for each voxel. **Right**: Measured metabolite peaks. As can be seen, Glx3 measures are elevated compared to Glx4, attributable to 2-HG accumulation in the tumor.

**Figure 2 diagnostics-13-02791-f002:**
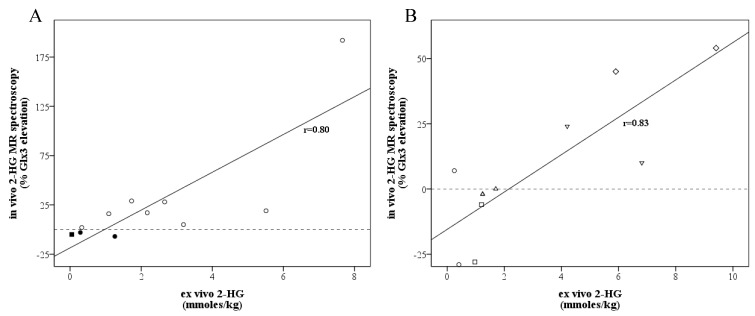
Scatter plot for correlations between in vivo 2-HG MR spectroscopy values (y-axis, measured as the percentage elevation of the Glx3 peak, the sum of 2-HG and Glx, compared to the Glx4 peak and the Glx3/4 ratio contralaterally) and ex vivo 2-HG measures (x-axis, measured in mmoles/kg tissue). (**A**) Correlations between in vivo 2-HG MR spectroscopy values and mean ex vivo 2-HG measures for all patients included. Additionally, 2-HG MR spectroscopy was negative in three patients (patients 7, 9, 10), indicated in black. Patient 9, indicated with the square, had an IDH wild-type tumor and a negative ex vivo 2-HG measure, whilst patients 7 and 10 were false negative. Circles were used for IDH-1 tumors, a square for the IDH wild-type case. (**B**) Correlations between voxel-specific in vivo 2-HG MR spectroscopy values and the 10 biopsy-matched ex vivo 2-HG measures among 5 patients. Two samples were taken per patient, indicated by the same symbol.

**Figure 3 diagnostics-13-02791-f003:**
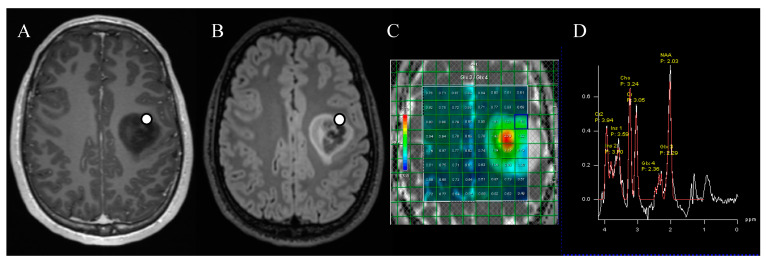
Marked location of tissue collection for ex vivo 2-HG concentration measurements in axial post-contrast T1-weighted (**A**) and FLAIR (**B**) sequences. The patient had a left frontal diffuse glioma, which turned out to be IDH1-mutated. The 2-HG MR spectroscopy (**C**) was positive, as demonstrated in the metabolite peaks (**D**) by an elevation of the Glx3 peak compared to the Glx4 peak, attributable to 2-HG.

**Table 1 diagnostics-13-02791-t001:** General characteristics and 2-HG results.

Pt	Age	Sex	IDH Mut	Histology (Grade)	Tumor Location (L/R)	2-HG MRS (%)	Ex Vivo [2-HG]	Ex Vivo 2-HG/ MRS Ratio	Predictive Value MRS
1	60	F	R132H	DA (II)	Temporal (L)	29	1.73	0.060	TP
2	32	M	R132C	DA (II)	Frontal (R)	5	3.19	0.638	TP
3	54	F	R132H	DA (II)	Frontal (R)	28	2.66	0.095	TP
4	48	M	R132H	DA (II)	Frontal (R)	17	2.17	0.128	TP
5	30	M	R132H	DA (II)	Midline	16	1.09	0.068	TP
6	34	M	R132H	DA (II)	Temporo-occipital (L)	2	0.33	0.165	TP
7	30	F	R132H	DA* (IV)	Fronto-parietal (R)	−3	0.29	−0.097	FN
8	24	M	R132S	DA (II)	Frontal (L)	192	7.66	0.040	TP
9	56	M	WT	GBM (IV)	Fronto-temporal (L)	−2	0.05	−0.025	TN
10	25	M	R132H	DA (II)	Frontal (L)	−7	1.38	−0.197	FN
11	27	F	R132C	DA (II)	Frontal (L)	19	5.51	0.290	TP

* Astrocytoma with necrosis and vascular proliferation. Abbreviations: 2-HG = 2-hydroxyglutarate; DA = diffuse astrocytoma; F = female; FN = false negative; GBM = glioblastoma; IDH = isocitrate dehydrogenase; L = left; M = male; MRS = magnetic resonance spectroscopy; Mut = mutation; Pt = patient; R = right; TN = true negative; TP = true positive; WT = wild type. 2-HG measures as determined by in vivo MR spectroscopy.

## Data Availability

The data presented in this study are available on request from the corresponding author.

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
