# Peer review of "The Correlation of In Vivo MR Spectroscopy and Ex Vivo 2-Hydroxyglutarate Concentration for the Prediction of Isocitrate Dehydrogenase Mutation Status in Diffuse Glioma"

_diagnostics, 2023, doi:10.3390/diagnostics13172791_

Round 1
Reviewer 1 Report
Excellent and well-thought-out study and manuscript. I have a few small questions/suggestions as follows:
Introduction:
- It would be appropriate to provide WHO reference # 2 at the end of paragraph 1.
- Line 47: provides references 1,4. Did the authors mean 1,3? (reference 4 from 2005 focused on TMZ+RT vs RT in GBM, and not on distinguishing prognosis of IDH mutant vs wt gliomas).
- Line 49: Again mentions reference 4. Did the authors mean 3? (see reason above)
Methods:
- Were oligodendrogliomas excluded from the study? If so, please mention in exclusion criteria.
Results:
- Error: In table 1, patient 10 in column 5 (histology, grade) states “GBM (II)”. Please check and correct it, e.g. to DA (II) if that is the case.
- Table 1, column 8: provides the values of “ex vivo [2- HG]”. However, in discussion, line 243 states “Our method avoids usage of a cut-off value, but rather measures the percentage elevation of Glx3 as compared to the Glx4 peak, making it more applicable to daily clinical practice”. I am not sure how clear it is from figure 2. Can the authors please also provide this ratio for each patient in a new column in table 1 or preferably in a separate table?
- Line 194 states “There was a strong correlation (r=0.80, p=0.003) between in vivo 2-HG MR spectroscopy and the mean ex vivo 2-HG concentration”. Is it possible for authors to also provide the ratio of ex vivo 2- HG and 2-HG MRS for each patient in a new column in table 1 or preferably in a separate table?
- Table 1: It appears to me that the values of ex vivo 2- HG and possibly 2-HG MRS of IDH mutations that are not R132H (i.e. R132C and R132S) are generally higher than those for R132H in IDH-mutant glioma cases. It would be interesting to compare the values of ex vivo 2- HG as well as above ratio in R132H (7 cases) versus non-R132H (3 cases) types of IDH-mutant gliomas, with statistical analysis for any significant difference (albeit small number).
- Line 165: For clarification, it may be helpful to include “(patients 7 and 10 in table 1)” at the end of the line “For two patients with a negative MR spectroscopy for 2-HG, histology did demonstrate an IDH1 mutation.”
Discussion:
- If the above results of ex vivo 2- HG and ratio of ex vivo 2- HG and 2-HG MRS amongst R132H (7 cases) versus non- R132H (3 cases) types of IDH-mutant gliomas are found to be valuable or statistically significant, it would be helpful to analyze those results with literature review in the discussion, and possibly briefly mention in the abstract as well.
- False negative 2-HG MRS in 2/10 IDH-mutant glioma patients (#7 and 10), i.e. 20% appears large enough even for a small study sample, and may be included in limitations of the study.
Thank you.
Author Response
Authors’ response to reviewers – Van Dijken et al. (Diagnostics-2552733)
Reviewer 1
Comment:
Excellent and well-thought-out study and manuscript. I have a few small questions/suggestions as follows:
Response:
We would like to thank the reviewer for the acknowledgement of our work and for critically reading our manuscript. We feel the suggestions made by the reviewer have strengthened our manuscript.
Comment:
Introduction: It would be appropriate to provide WHO reference # 2 at the end of paragraph 1.
Response:
In line with the reviewer’s suggestion, we have added reference 2 to the end of paragraph 1.
Comment:
Line 47: provides references 1,4. Did the authors mean 1,3? (reference 4 from 2005 focused on TMZ+RT vs RT in GBM, and not on distinguishing prognosis of IDH mutant vs wt gliomas).
Line 49: Again mentions reference 4. Did the authors mean 3? (see reason above)
Response:
We thank the reviewer for pointing this out. Indeed reference 3 should have been mentioned here instead of reference 4, and we corrected this accordingly. In line 49 we also added reference 3 to the sentence, but kept reference 4 in place as well, since the Stupp regimen is currently still the standard treatment for IDH wild type GBM.
Comment:
Methods: Were oligodendrogliomas excluded from the study? If so, please mention in exclusion criteria.
Response:
Oligodendrogliomas were indeed excluded from this study. We have added this to the methods section.
Comment:
Results: Error: In table 1, patient 10 in column 5 (histology, grade) states “GBM (II)”. Please check and correct it, e.g. to DA (II) if that is the case.
Response:
We thank the reviewer for the alertness of pointing this error out. Indeed patient 10 had an IDH mutated diffuse astrocytoma (grade II). We changed this accordingly.
Comment:
Table 1, column 8: provides the values of “ex vivo [2- HG]”. However, in discussion, line 243 states “Our method avoids usage of a cut-off value, but rather measures the percentage elevation of Glx3 as compared to the Glx4 peak, making it more applicable to daily clinical practice”. I am not sure how clear it is from figure 2. Can the authors please also provide this ratio for each patient in a new column in table 1 or preferably in a separate table?
Response:
We agree with the reviewer that this phrasing was somewhat unclear. The 2-HG MRS values were calculated as a percentage of Glx3 elevation compared to Glx4 (paragraph 3 of methods section 2.3. MR spectroscopy acquisition, from line 125), rather than using an absolute and predefined cut-off value as other studies have used.
To clarify this, we have chanced this to: “Our method avoids usage of a predefined absolute cut-off value, but rather measures the percentage elevation of Glx3 as compared to the Glx4 peak, making it more applicable to daily clinical practice since values >0% indicate an higher Glx3 concentration than Glx4 due to 2-HG.”
We have also added “(%)” to the 2-HG MRS column to further clarify that the MRS results were measured as a percentage of Glx3 elevation compared to Glx4, as specified in the method section.
Comment:
Line 194 states “There was a strong correlation (r=0.80, p=0.003) between in vivo 2-HG MR spectroscopy and the mean ex vivo 2-HG concentration”. Is it possible for authors to also provide the ratio of ex vivo 2- HG and 2-HG MRS for each patient in a new column in table 1 or preferably in a separate table?
Response:
We have incorporated the ex vivo 2-HG/2-HG MRS ratios as an additional column in table 1 as suggested by the reviewer.
Comment:
Table 1: It appears to me that the values of ex vivo 2- HG and possibly 2-HG MRS of IDH mutations that are not R132H (i.e. R132C and R132S) are generally higher than those for R132H in IDH-mutant glioma cases. It would be interesting to compare the values of ex vivo 2- HG as well as above ratio in R132H (7 cases) versus non-R132H (3 cases) types of IDH-mutant gliomas, with statistical analysis for any significant difference (albeit small number).
Response:
The reviewer raises a very interesting question. Indeed, it seems that the 3 cases of non-R132H IDH mutations have a higher concentration of ex vivo 2-HG. Again, due to the limited number of patients, these findings should be treated with caution. As suggested, we compared the mean ex vivo 2-HG concentration between R132H cases and non-R132H cases, which was found to differ significantly. Additionally, we also compared the newly added ratios, which again, demonstrated a statistically significant difference between the two groups. We added these results to the manuscript along with the descriptive information of the different IDH mutation types. The following sentences were added to the results section:
Seven patients had a tumor with the typical R132H mutation, whereas non-R132H mutations (R132C n=2, R132S n=1) were found in the three other patients with IDH1 mutated tumors.
The mean ex vivo 2-HG concentration was higher in non-R132H mutated cases (5.45 mmoles/kg) than in cases with a typical R132H IDH mutation (1.38 mmoles/kg), p=0.001. This was also the case for the ratios between the ex vivo 2-HG concentration and 2-HG spectroscopy (table 1), with non-R132H cases demonstrating higher ratios (p=0.035).
When excluding the three cases with a non-R132H mutation, the correlation did not change (r=0.81, p=0.014).
Comment:
Line 165: For clarification, it may be helpful to include “(patients 7 and 10 in table 1)” at the end of the line “For two patients with a negative MR spectroscopy for 2-HG, histology did demonstrate an IDH1 mutation.”
Response:
We thank the reviewer for this excellent suggestion and have included this at the end of the sentence.
Comment:
Discussion: If the above results of ex vivo 2- HG and ratio of ex vivo 2- HG and 2-HG MRS amongst R132H (7 cases) versus non- R132H (3 cases) types of IDH-mutant gliomas are found to be valuable or statistically significant, it would be helpful to analyze those results with literature review in the discussion, and possibly briefly mention in the abstract as well.
Response:
We have added a paragraph to further discuss the findings of higher 2-HG concentrations among non-R132H cases (see below). Interestingly, the observation by the reviewer has previously been studies and it was demonstrated by Pusch et al (included as new reference 12) that non-R132H subtypes generally have higher 2-HG formation. The others of that study hypothesize that the predominance of the R132H subtype could be due to the proliferative effect of a mildly elevated 2-HG concentration only (as produced by R132H types), where as higher concentrations might not be beneficial for tumor cells.
We feel that this extra analysis of the IDH1 subtypes and short literature review adds interesting information to our manuscript and we would like to thank the reviewer again for this observation.
In our cohort, three out of 10 patients had a rare non-R132H IDH1 mutation, which is more than expected. It is known that >90% of IDH1 mutant gliomas harbor a R132H mutation, with approximately 4% harboring a R132C mutation and the other subtypes occurring even less frequently [12]. The three non-R132H cases demonstrated a higher ex vivo 2-HG concentration than the R132H cases. However, when we excluded the non-R132H cases, our results did not change with a high correlation between 2-HG MR spectroscopy and the ex vivo 2HG concentration. The higher concentration of 2-HG for non-R132H subtypes has also been described by an earlier study [12]. This study by Pusch et al hypothesized that only a moderate increased 2-HG concentration, as produced by R132H mutations, had beneficial proliferating effects and may explain the predominance of R132H cases [12]. The implications of different IDH1 mutation subtypes on the diag-nostic performance of 2-HG MR spectroscopy should be further studied in larger cohorts.
Comment:
False negative 2-HG MRS in 2/10 IDH-mutant glioma patients (#7 and 10), i.e. 20% appears large enough even for a small study sample, and may be included in limitations of the study.
Response:
We have added two sentences to the re-written limitations section, including the 20% false negative rate of 2-HG MRS in our cohort, which seems unproportionally high. To stress the limited generalizability of our small study (as also mentioned by this reviewer), we have added the following two sentences:
Despite our positive findings, two IDH1 mutated gliomas had false negative 2-HG MR spectroscopy results (20%), which is higher than previously described. The limited generalizability is also highlighted by the relatively large number of non-R132H mutations.
=========================================
Reviewer 2 Report
The manuscript presents a study focused on the correlation between in vivo MR spectroscopy-derived 2-hydroxyglutarate (2-HG) measurements and ex vivo 2-HG concentration in resected tumor tissue for the prediction of isocitrate dehydrogenase (IDH) mutation status in diffuse glioma. The proposed method offers a noninvasive approach for determining IDH mutation status. The study demonstrates a strong correlation between in vivo 2-HG MR spectroscopy and ex vivo 2-HG concentration, primarily in patients with lower-grade gliomas. The manuscript provides important insights into the potential application of this technique in glioma diagnosis and subtype classification.
The manuscript titled "Correlation of in vivo MR spectroscopy and ex vivo 2-hydroxyglutarate concentration for prediction of isocitrate dehydrogenase mutation status in diffuse glioma" by Bart R.J. van Dijken addresses a significant aspect of glioma diagnosis and prognosis using a promising noninvasive method. Although, the reviewer commends the findings, particularly the strong correlation between in vivo 2-HG MR spectroscopy and ex vivo 2-HG concentration, are promising and have potential clinical implications, a few minor comments are suggested for further enhancement and clarity.
Minor Comments:
Limitations Section: The reviewer suggests adding a separate "Limitations" section at the end of the Conclusion. Given the small sample size of the study, the limitations section would help clarify the potential constraints of drawing definitive conclusions from a limited dataset. Furthermore, the small sample size might limit the study's generalizability, and the inclusion of this limitation would provide a comprehensive view of the study's implications.
Voxel-Specific MR Spectroscopy: The manuscript mentions that voxel-specific MR spectroscopy comparison with ex vivo 2-HG concentration was available in a subgroup of patients (5 patients with 10 samples). It would be beneficial to briefly explain the reasons behind this limited availability and discuss how this subgroup was representative of the larger cohort, considering its potential impact on the study's outcomes.
Author Response
Authors’ response to reviewers – Van Dijken et al. (Diagnostics-2552733)
Reviewer 2
Comment:
The manuscript presents a study focused on the correlation between in vivo MR spectroscopy-derived 2-hydroxyglutarate (2-HG) measurements and ex vivo 2-HG concentration in resected tumor tissue for the prediction of isocitrate dehydrogenase (IDH) mutation status in diffuse glioma. The proposed method offers a noninvasive approach for determining IDH mutation status. The study demonstrates a strong correlation between in vivo 2-HG MR spectroscopy and ex vivo 2-HG concentration, primarily in patients with lower-grade gliomas. The manuscript provides important insights into the potential application of this technique in glioma diagnosis and subtype classification.
The manuscript titled "Correlation of in vivo MR spectroscopy and ex vivo 2-hydroxyglutarate concentration for prediction of isocitrate dehydrogenase mutation status in diffuse glioma" by Bart R.J. van Dijken addresses a significant aspect of glioma diagnosis and prognosis using a promising noninvasive method. Although, the reviewer commends the findings, particularly the strong correlation between in vivo 2-HG MR spectroscopy and ex vivo 2-HG concentration, are promising and have potential clinical implications, a few minor comments are suggested for further enhancement and clarity.
Response:
We thank the reviewer for the extensive review. Reliable noninvasive determination of IDH mutation status indeed is of particular interest and potentially has many clinical implications, ranging from diagnosing to treatment follow-up. Our study uses a straight forward method, based on the elevation of Glx3 (due to 2-HG) compared to Glx 4. We demonstrated a strong correlation between our findings and the ex vivo 2-HG concentration in surgically removed tumor tissue. To our knowledge, this is the first study demonstrating such results and albeit a small study, should prompt future studies to validate our results in larger cohorts.
We would like to thank the reviewer for the kind words and for providing the suggestions below, which we have addressed in the new version of our manuscript and have indeed improved it.
Comment
Limitations Section: The reviewer suggests adding a separate "Limitations" section at the end of the Conclusion. Given the small sample size of the study, the limitations section would help clarify the potential constraints of drawing definitive conclusions from a limited dataset. Furthermore, the small sample size might limit the study's generalizability, and the inclusion of this limitation would provide a comprehensive view of the study's implications.
Response:
Our straightforward methodology for 2-HG spectroscopy could potentially enhance the noninvasive determination of IDH mutation status in gliomas. Our study is also the first to directly compare 2-HG spectroscopy with ex vivo measurements. Nevertheless, the small sample size is indeed the largest limitation of our study, as correctly mentioned by the reviewer.
In line with the reviewer’s suggestion we have clarified the limitation section and the limited generalizability of our results, by including it as a separate section. Furthermore, we have also added two sentences to the conclusion section: “Despite the small number of patients included in this study, with a strong correlation between 2-HG MR spectroscopy with the ex vivo 2-HG concentration was found. Our results should prompt future studies to validate our findings in larger cohorts. noninvasive approach for determining IDH mutation status.”
Comment:
Voxel-Specific MR Spectroscopy: The manuscript mentions that voxel-specific MR spectroscopy comparison with ex vivo 2-HG concentration was available in a subgroup of patients (5 patients with 10 samples). It would be beneficial to briefly explain the reasons behind this limited availability and discuss how this subgroup was representative of the larger cohort, considering its potential impact on the study's outcomes.
Response:
Unfortunately, the biopsy location was only stored for 5 patients of our cohort, due to technical issues with the neuronavigation software. This, however, was only discovered at the end of the study. Therefore, there was no specific selection of patients for this procedure and we have no concerns about the representation of this subgroup compared to the entire cohort.
We agree that the reason why biopsy data was only known for 5 patients should be better specified in the manuscript. Therefore, we added this information to the methods section: “Unfortunately, due to technical issues with the neuronavigation system, the biopsy lo-cations for the other patients was not stored”). A similar sentence was added to the limitation section at the end of the manuscript.